# An Innovative Chiral UPLC-MS/MS Method for Enantioselective Determination and Dissipation in Soil of Fenpropidin Enantiomers

**DOI:** 10.3390/molecules27196530

**Published:** 2022-10-02

**Authors:** Rui Li, Yanqing Zhang, Yanhong Li, Zihao Chen, Zhen Wang, Minghua Wang

**Affiliations:** Department of Pesticide Science, College of Plant Protection, Nanjing Agricultural University, State & Local Joint Engineering Research Center of Green Pesticide Invention and Application, Nanjing 210095, China

**Keywords:** fenpropidin, enantio-separation, absolute configuration, Box-Behnken design, enantioselective dissipation, molecular docking

## Abstract

As a chiral piperidine fungicide, fenpropidin has been widely used to control plant diseases. However, there are rare studies that have investigated fenpropidin at the enantiomer level. In this study, the single-factor analysis combined with a Box-Behnken design was used to obtain the optimal enantio-separation parameters of the fenpropidin enantiomers on ultra-performance liquid chromatography-tandem mass spectrometry. The absolute configuration of two fenpropidin enantiomers was confirmed for the first time using electron circular dichroism and optical activity. On the Lux cellulose-3 column, *S*-(-)-fenpropidin flowed out before *R*-(+)-fenpropidin. The enantio-separation mechanism was revealed by molecular docking. A modified QuEChERS method was developed for the trace determination of the fenpropidin enantiomers in seven food and environmental substrates. The average recoveries were 71.5–106.1% with the intra-day and inter-day relative standard deviations of 0.3–8.9% and 0.5–8.0%. The method was successfully verified by enantioselective dissipation of fenpropidin in soil under the field. *R*-(+)-fenpropidin dissipated faster than *S*-(-)-fenpropidin, and the half-lives were 19.8 d and 22.4 d. This study established a brand-new effective chiral analysis method for the fenpropidin enantiomers, providing a basis for accurate residue monitoring and the risk assessment of fenpropidin.

## 1. Introduction

Pesticides play an important role in controlling pests, diseases and weeds, protecting crops and regulating growth. It is worth noting that the number of chiral pesticides is relatively large, accounting for 30% of all pesticides sold worldwide, and it is increasing gradually with the introduction of complex structures [1,2]. The chiral pesticides usually contain more than two enantiomers. The enantiomers of each chiral pesticide usually have the same physical and chemical properties; however, they have different recognition sites for organisms, which may exhibit different biological activity, non-target toxicity and environmental behavior [3,4,5,6]. Therefore, studies on the racemic levels of chiral pesticides may obtain inaccurate data. Enantioselectivity of chiral pesticides has received considerable attention worldwide. For instance, Li et al. found that the fungicidal activity of *S*-(+)-fluxametamide is up to 52.1–304.4-fold higher than *R*-(-)-fluxametamide [7]. Li reported imazamox has enantioselective toxicity to *L**emna minor*, *R*-imazamox was about six times more toxic to *Lemna minor* than *S*-imazamox [8]. Wang found that pydiflumetofen had enantioselective degradation in soil, *R*-(+)-enantiomer degraded faster than *S*-(-)-enantiomer [9]. In order to improve the efficiency and reduce the use of chiral pesticides, further avoid the threat of inefficient and highly toxic enantiomers to the environment and non-target organisms, some countries, such as the Netherlands and Switzerland, have registered single isomers [10]. Therefore, enantioselective studies on chiral pesticides can obtain more scientific, reasonable and accurate data for a comprehensive risk assessment of chiral pesticides.

Fenpropidin, 1-[(*RS*)-3-(4-tert-butylphenyl)-2-methylpropyl] piperidine, is a typical chiral pesticide developed by Syngenta Co., Ltd. and belongs to the new piperidine fungicides which contain a chiral center and two enantiomers. Fenpropidin eventually inhibits sterol synthesis by inhibiting Δ^14^-reductase and Δ^8^ → Δ^7^-isomerase [11,12]. Maximum residue limits (MRLs) for fenpropidin have been established in many countries. The MRLs of fenpropidin in wheat were 1 mg/kg in China, 10 mg/kg in bananas in America and Canada and in the European Union, the MRLs were 0.01 to 0.6 mg/kg in different foods. However, most studies of fenpropidin were still carried out at the racemic level. For example, LC-MS/MS and GC-MS were used by Zhao et al. to study the dissipation of fenpropidin in wheat and soil and found that the half-life was between 3.1 to 3.3 days in wheat plants and 13.4 to 16.5 days in soil [13]. Unfortunately, fenpropidin has been found in environmental substrates. Hvězdova et al. tested different types of pesticides in a variety of soils and found that 20% of soils contained fenpropidin and 13% contained fenpropidin at levels greater than 0.01 mg/kg [14]. Schafer detected fenpropidin in 16 small streams, and its concentration was as high as 1 mg/kg [15]. Residual fenpropidin may pose a threat to environmental safety and human health; thus, accurate monitoring of fenpropidin in agricultural products and environmental samples to ensure food and environmental safety is necessary. However, as far as we know, there are few reports on the analysis of fenpropidin from the enantiomer level. Only Buerge et al. reported the stereoselective metabolism of fenpropidin in beet and wheat by GC-MS/MS [16]. Therefore, establishment of a rapid, sensitive and accurate chiral analysis method for the detection of the fenpropidin enantiomers in food and environmental samples is an urgent task.

Herein, a novel, sensitive and efficient chiral analysis method for the determination of fenpropidin enantiomers in bananas, grapes, apples, wheat, soybeans, rice and soil was established by ultra-performance liquid chromatography-tandem triple quadrupole mass spectrometry (UPLC-MS/MS). A novel Box-Behnken design (BBD) experiment was performed to comprehensively and systematically evaluate the influence of chiral separation parameters (chiral stationary phases (CSPs), mobile phase composition and proportion, temperature, flow rate) for the separation on the basis of single factor optimization test. The absolute configuration of the fenpropidin enantiomers was confirmed by electron circular dichroism (ECD) and polarimetry. Furthermore, the chiral recognition mechanism of the fenpropidin enantiomers was studied by molecular docking under ideal conditions. The established method was successfully performed on the enantioselective dissipation of the fenpropidin enantiomers in soil. Simultaneously, the stability of the fenpropidin enantiomers in different solvents and water was studied. These results will provide more accurate and reliable data for environmental monitoring and risk assessment of fenpropidin.

## 2. Results and Discussion

### 2.1. Enantio-separation and Optimization

#### 2.1.1. Optimization of Chiral Stationary Phases

CSP is the dominant factor in chiral recognition [17,18]. The polysaccharide-based chiral column is one of the most widely used chiral columns [19,20]. Thus, the enantio-separation effect of fenpropidin on three different cellulose-derived CSPs (Lux cellulose-1 (L1), Lux cellulose-2 (L2) and Lux cellulose-3 (L3)) was studied. We have studied the enantio-separation of fenpropidin on different chiral columns under various mobile phases such as formic acid water/acetonitrile (methanol), ammonia solution/acetonitrile (methanol), ammonium acetate/acetonitrile (methanol) and ammonium formate/acetonitrile (methanol). Fenpropidin enantiomers on the chiral column L1 and L2 failed to achieve complete baseline separation. But complete baseline separation of the fenpropidin enantiomers was obtained on the L3 (Figure 1a). Therefore, L3 was selected as the stationary phase in this study to separate the fenpropidin enantiomers.

#### 2.1.2. Optimization of Mobile Phase

The retention time (Rt), elution and separation of chiral compounds are affected by the mobile phase [21]. The common organic phases (methanol and acetonitrile) combined with several aqueous phases (formic acid water, ammonia solution, ammonium acetate and ammonium formate) were used to study the enantio-separation effect of fenpropidin. Acetonitrile as the organic phase coupled with different aqueous phases could not achieve enantio-separation, while methanol can achieve a good separation effect. Typical chromatograms were shown in Figure 1b.

What is more, methanol combined with formic acid, ammonium acetate, and ammonium formate could not achieve the baseline separation. Only a methanol/ammonia solution could achieve the enantio-separation of fenpropidin (Figure 1c). That is probably because an appropriate buffer solution can significantly change the ionization effect, adjust the pH and improve the peak shape and signal response [22]. Fenpropidin is alkaline, and ammonia solution can promote its ionization. Therefore, the ammonia solution was the best buffer for the enantio-separation of fenpropidin. In addition, the influence of 0.05%, 0.1% and 0.2% ammonia solution on the mobile phase was compared (Figure 1d). All of them could successfully separate the fenpropidin enantiomers with the resolution (Rs) of 1.80, 1.96 and 2.27, respectively. When the mobile phase was 0.1% ammonia solution, the maximum response value was obtained (Appendix A). Considering the Rs, response value and the degree of acid and alkali tolerance of a chiral column, 0.1% ammonia solution was selected as the mobile phase.

#### 2.1.3. Effects of the Column Temperature on Enantio-Separation

The adsorption and desorption rates of target compounds in stationary phases usually were affected by temperature, thus, affecting the separation effect [23]. The van’t Hoff equations got good linear relationships (R^2^ > 0.9941) at the column temperature between 20–40 °C. The Appendix A showed the detailed thermodynamic parameters. The Rs of the enantiomers gradually decreased when the temperature increased (Appendix A). In addition, the ∆∆H^0^ and ∆∆S^0^ were −2.97 kJ/mol and −6.38 J/(mol·k) which indicated enthalpy drove the separation of the fenpropidin enantiomers.

### 2.2. Box-Benhnken Design

The interaction of many factors affects the enantio-separation of chiral compounds. Considering from the single factor level is not comprehensive enough, BBD can predict the main influence variables and the interaction of multiple variables [24]. It is usually used to evaluate the nonlinearity relationship of three to seven input variables and output variables. To further study the influence of Rt (Y_1_) and Rs (Y_2_) of the fenpropidin enantiomers by flow rate (X_1_), the percentage of methanol (X_2_), and temperature (X_3_), a BBD was used in this study. In each of these experiments, two dependent variables changed and the other one remained at the medium level. The accuracy and precision of the BBD were proven through the dependent variable experiment at the center point with five repetitions. Two second-order models (Equations (1) and (2)) eliminating insignificant factors were obtained by the Design Expert 8.0.6 trial software.
(1)$$Y_{1}=16.69-2{.61X}_{1}+12{.16X}_{2}-1{.30X}_{3}+0{.1150X}_{1}X_{3}+0{.7714X}_{1}{}^{2}+0{.3114X}_{3}{}^{2}$$
(2)$$Y_{2}=2.24-0.0977X_{1}+0.9115X_{2}+0.0015X_{3}-0.0879X_{3}{}^{2}$$

The rationality of the fitted models (Y_1_ and Y_2_) was evaluated by the ANOVA of Rt and Rs (Appendix A). In this study, a good linearity was obtained with R^2^ = 0.9581 and R^2^ = 0.9625 for Y1 and Y2. The two fitted models were significant differences (*p* < 0.0001). The significant factor affecting Rt was methanol percentage (X_2_) (*p* < 0.0001). The percentage of methanol (X_2_) affected the Rs (*p* < 0.0001).

As shown in Figure 2, the interaction of multiple factors in the chiral separation process was illustrated by the three-dimensional (3D) response surface plots and two-dimensional (2D) contour plots. The optimum conditions were obtained through 17 BBD experiments with the flow rate of 0.8 mL/min, percentage of methanol of 88.3%, and temperature of 32.7 °C. The predicted results by the BBD were in perfect agreement with the actual results on UPLC-MS/MS, with the Rt of the first enantiomer being 11.00 min and the Rs being 1.96.

### 2.3. Absolute Configuration

The specific optical rotations of peak 1 and peak 2 were calculated as [α] = −9.57° and [α] = +9.89° (methanol, c = 0.01). The calculated ECD spectrum was acquired by using the lowest energy conformation of each fenpropidin enantiomer (Figure 3). It was found that the calculated and experimental ECD spectra were consistent, and there was a mirror symmetry phenomenon. Thus, peak 1 was *S*-(-)-fenpropidin and peak 2 was *R*-(+)-fenpropidin (Figure 4).

### 2.4. Molecule Docking

The binding modes and interaction between the fenpropidin enantiomers and CSPs were investigated by molecular docking to reveal the enantio-separation mechanism of the fenpropidin enantiomers.

The enantio-separation of fenpropidin was carried out based on CSPs. Only the L3 achieved the baseline separation of the fenpropidin enantiomers, while the L1 and L2 did not achieve the baseline separation at all. Therefore, the possible interaction between chiral compounds and CSPs can be explored by molecular docking. And the binding modes of the fenpropidin enantiomers and the L3 were presented in Figure 5a,b. The active pocket consisted of several glucose derivatives, and the fenpropidin enantiomers interacted with the L3 including the π-cation interaction and hydrogen-bond interaction. As shown in the Figure 5a,b, although two enantiomers had the same type of interaction with the L3, the distance of the hydrogen bond and π-cation between two enantiomers and the binding site was different due to the different stereostructure of the fenpropidin enantiomers. The heterocyclic N atom in two fenpropidin enantiomers formed hydrogen bonds with the O-H group bonds on the L3 branch, and combined with the aromatic ring to form π-cation interaction. The distances between the two atoms of the hydrogen bond formed by *R*-enantiomer and *S*-enantiomer with the binding site of the L3 were 2.17 Å and 2.39 Å, respectively. In addition, fenpropidin had π-cation interaction with two π systems in the active center pocket. The difference in spatial structure between *R*-fenpropidin and *S*-fenpropidin led to the different distances of the π system and the cations on the ligand. Compared with some classical interactions (such as hydrogen-bond, electrostatic and hydrophobic interactions), π-cation interaction was considered to be a new type of intermolecular interaction. Such interaction had characteristic requirements for the binding geometry [25]. The strength of the π-cation interaction is comparable to hydrogen bonds. As shown in Figure 5a,b, the distances of π-interaction between the *R*-enantiomer and the L3 were 4.36 Å and 5.32 Å, and the distances between *S*-enantiomer and the L3 were 4.26 Å and 6.22 Å. The geometry of one π-cation force in *S*-enantiomer was greater than 6 Å. Therefore, although the fenpropidin enantiomers have the same interaction force as the L3, the distances between two enantiomers and the L3 were different due to the different spatial structures. As shown in the Appendix A, the results of free-binding energy were consistent with the separation results of the fenpropidin enantiomers on the L3. The binding energies between *S*-enantiomer and *R*-enantiomer were significantly different. The analysis based on the binding energy and intermolecular interaction showed that the binding ability of *S*-fenpropidin to the L3 was weaker than that of *R*-fenpropidin, so *S*-fenpropidin flowed out before *R*-fenpropidin on the L3 column. The molecular docking results can clearly explain the mechanism of enantio-separation of fenpropidin. It is clear that the mechanism of fenpropidin enantio-separation on the L3 may be due to the difference in the distances of the hydrogen bond and π-cation interaction between the fenpropidin enantiomers and the L3.

As shown in Appendix A, *R*-enantiomer and *S*-enantiomer formed π-cation interactions and hydrogen bonds with the L1, respectively. *R*-enantiomer and the L2 formed hydrogen bonds, and *S*-enantiomer formed π-cation interactions as shown in Appendix A. Based on the previous results, fenpropidin enantiomers had similar intermolecular forces with the L1 and L2, and their free-binding energies were not significantly different. Therefore, we speculated that only containing hydrogen bonds or π-cation interactions was not enough to achieve enantio-separation. The difference in the free-binding energy between CSPs and two enantiomers also showed that there had no significant difference on the L1 and L2 (Appendix A). Therefore, the enantio-separation of fenpropidin was only achieved on the L3. The results of molecular docking strongly confirmed the results of the optimization experiment of enantio-separation and explained the mechanism of fenpropidin enantio-separation.

### 2.5. Method Validation

#### 2.5.1. Specificity, Linearity, LOQs and Matrix Effect

This method was highly specific, and there was no interfering substance in the presence of the fenpropidin enantiomers during Rt in all blank samples.

An excellent linearity of the fenpropidin enantiomers in solvent and seven matrix-matched calibration curves was obtained with R^2^ ≥ 0.9988 under the concentration of 5–500 μg/kg (Table 1). The limit of quantifications (LOQs) of the fenpropidin enantiomers were 5 μg/kg.

The fenpropidin enantiomers showed no matrix effect (ME) in wheat, apples, soybeans and soil, but moderate ME with signal suppression in grapes and bananas. There was no ME in rice with *S*-(-)-fenpropidin, but there was a matrix suppression effect in rice with *R*-(+)-fenpropidin. ME may be due to coextract which may significantly interfere with the analysis process of target compounds and affect the accuracy of analysis results [26,27]. Therefore, the standard calibration curve of matrix matching was used for accurate quantification

#### 2.5.2. Accuracy and Precision

The mean recoveries of the fenpropidin enantiomers in seven matrixes were 71.5–106.1%, the intraday and intraday relative standard deviations (RSDs) were 0.3–8.9% and 0.5–8.0% (Table 2). The data showed that the enantioselective detection method of the fenpropidin enantiomers in wheat, apples, soybeans, grapes, bananas, rice and soil had good accuracy and precision.

### 2.6. Stability of Fenpropidin Enantiomers

It has been proved that the configuration of chiral pesticide enantiomers is unstable under experimental conditions and is prone to configuration transformation [28,29]. The stability of the fenpropidin enantiomers dissolved in different solvents was verified. And the results showed that there was no configuration transformation that occurred in the fenpropidin enantiomers.

### 2.7. Enantioselective Dissipation of Fenpropidin in Soil

The kinetic equations for the dissipation of *S*-(-)-fenpropidin and *R*-(+)-fenpropidin in soil were C = 0.7882e^−0.031t^ (R^2^ = 0.8728) and C = 0.8624e^−0.035t^ (R^2^ = 0.9121), which well fitted the first order kinetic equation. The half-lives of *R*-(+)-fenpropidin and *S*-(-)-fenpropidin in soil were 19.8 d and 22.4 d, the dissipation of the fenpropidin enantiomers in soil was enantioselective (*p* < 0.05) (Appendix A), and *R*-(+)-fenpropidin dissipated faster than *S*-(-)-fenpropidin (Appendix A). EF at 2 h and 35 d were 0.48 and 0.52 (Appendix A). This method can be helpful to improve the accuracy of the risk assessment of fenpropidin.

## 3. Materials and method

### 3.1. Reagents and Materials

The fenpropidin (≥95%) was obtained from Dr. Ehrenstorfer (Augsburg, Germany). Two fenpropidin enantiomers (≥98% purity) were acquired from Chiralway Biotech Co., Ltd. (Shanghai, China). HPLC-grade ammonia solution was obtained from Shanghai Aladdin Biotech Co., Ltd. (Shanghai, China). HPLC-grade methanol and acetonitrile were bought from Germany Merck KGaA (Darmstadt, Germany). Ultra-pure water was purchased from China Resources C’estbon Beverage (China) Co. Ltd. (Shenzhen, China). Primary secondary amine (PSA, 40–63 μm) was purchased from ANPEL Laboratory Technologies, Inc. (Shanghai, China). Racemic fenpropidin and its two enantiomers were dissolved in HPLC-grade methanol to prepare stock standard solutions and all of them were placed in the dark at −20 °C.

### 3.2. Instrumental Analysis

A Waters ACQUITY UHPLC system tandem triple quadrupole mass spectrometer (Waters Corp., Milford, MA, USA) was used for the enantio-separation and analysis of fenpropidin in a positive electrospray ionization source (ESI+). The L3 chiral column (250 × 4.6 mm inner diameter, 5 μm, Phenomenex, Torrance, CA, USA) was selected for separating the fenpropidin enantiomers. The flow rate and volume ratio of the mobile phase (methanol and 0.1% ammonia solution) were 0.8 mL/min and 88.3:11.7. And the column temperature was 32.7 °C. Multiple-reaction monitoring mode was used to perform MS analysis. The mass conditions were set as follows: 1.25 kV capillary voltage, 500 °C desolvation temperature, 150 °C source temperature. A 50 L/h cone gas flow (99.95% nitrogen) and 1000 L/h desolvation gas flow (99.95% nitrogen) were used. The collision gas was 99.99% argon at a pressure of 2 × 10^−3^ mbar in the T-wave cell. The quantitative and qualitative determination of the fenpropidin enantiomers were through the characteristic product ions emerging from hydrogen adduct [M+H]^+^ (*m/z* 274 > 147) and (*m/z* 274 > 132) under the cone voltage and the collision energy of 26 and 40 V. The data was collected and analyzed by Masslynx NT version 4.2 (Waters, Milford, MA, USA).

### 3.3. Separation Condition Optimization

In the case of comprehensively and systematically assessing the effect of separation parameters on the enantio-separation of the fenpropidin enantiomers, the influence of various chiral columns and mobile phase composition was preliminarily assessed through the single factor analysis. A novel BBD experiment was applied to study the influence of multiple factors (column temperature, flow rate, mobile phase ratio) on enantio-separation from three levels (high, medium and low) with a total of 17 experiments by the Design Expert 8.0.6 trial software, and, finally, the optimal separation conditions were obtained. Specific conditions are as follows: the column temperature was between 25 and 35 °C, the methanol ratio was 80% to 95% and the flow rate was 0.6 mL/min to 0.8 mL/min. Response variables were set as the Rt of Peak 1 and the Rs between Peak 1 and Peak 2. The quadratic model was utilized to calculate the response surface as shown in Equation (3) [30]:(3)$$Y=b_{0}+b_{1}X_{1}+b_{2}X_{2}+b_{3}X_{3}+b_{11}X_{1}{}^{2}+b_{22}X_{2}{}^{2}+b_{33}X_{3}{}^{2}+b_{12}X_{1}X_{2}+b_{13}X_{1}X_{3}+b_{23}X_{2}X_{3}$$
where Y is the predicted response, bn are quadratic coefficients, X_1_, X_2_, and X_3_ is the flow rate, the percentage of methanol, and the column temperature respectively.

The effect of different optimization conditions on the separation was evaluated by the capacity factor (k), separation factor (α) and Rs using Equations (4), (5) and (6). The van’t Hoff equations were used to calculate the thermodynamic parameters as shown in Equations (5) and (6) [31].
(4)$$k=(t_{R}-t_{0})/t_{0}$$
(5)$$\alpha =k_{2}/k_{1}$$
(6)$$Rs=2(t_{2}-t_{1})/(w_{1}+w_{2})$$
(7)$$\ln k=-\Delta H^{{}_{0}}/RT+\Delta \Delta S^{0}/R+\ln\Phi$$
(8)$$\ln\alpha =-\Delta \Delta H^{0}/RT+\Delta \Delta S^{0}/R$$
where t_R_, t_0_ and w is the Rt, the void time, the peak width, respectively; R, T and Φ is the gas constant, the absolute temperature and the ratio of the solid phase and the mobile phase, respectively. ΔΔH^0^ and ΔΔS^0^ are the enthalpy and entropy variations.

### 3.4. Determination of Specific Optical Rotation

A SGW-1 polarimeter (Shanghai INESA Physico-Optical Instrument Co., Ltd., Shanghai, China) was utilized to determine the optical activity of each fenpropidin enantiomer (acetonitrile, 0.01 g/mL) at 589.4 nm, repeat three times. Using Equation (9) to calculate the specific rotation.
(9)[α]=α/(c×L)
where [α], α, C and L represent specific rotation, optical rotation, the concentration of the fenpropidin enantiomers (g/mL) and the width of the quartz tube (dm).

### 3.5. Confirmation of Absolute Configuration

In stereochemical analysis, ECD is generally considered to be one of the most effective techniques for confirming the absolute configuration of chiral compounds [32]. The experimental ECD spectra of the standard solution of two fenpropidin enantiomers (acetonitrile, 10^−5^ mol/mL) were conducted by the J815 circular dichroism spectropolarimeter (Jasco, Tokyo, Japan) at 25 °C. The standard solution should be placed in a 0.1 cm quartz cell. The measurement was carried out at the scanning wavelength of 200–400 nm and a scan speed of 50 nm/min. The experimental ECD spectra were drawn by Origin software (version 8.61).

At present, the method of quantum chemical calculations was usually used to confirm the absolute configurations of chiral compounds [33]. Gaussian 09 W software was used to acquire the calculated ECD spectra of the fenpropidin enantiomers. Firstly, the molecular mechanics field (MMFF94) was used to optimize the 3D structures of *R*-fenpropidin and *S*-fenpropidin. The geometric optimization and frequency calculation were conducted to get the most stable configuration of two enantiomers based on the B3LYP function of the 6-311+G (2d, p) basis set. The absolute configurations of the fenpropidin enantiomers were confirmed based on the similarity between the calculated ECD and the experimental ECD spectra.

### 3.6. Chiral Stationary Phase Recognition Mechanism

A computer simulation was implemented to explore the enantio-separation mechanism between two enantiomers and different CSPs. The 3D structures of the chiral stationary phases (L1, L2, L3) were derived based on Yamamoto’s research [34]. The process was performed using Schrodinger Maestro Suite 2020 which was professional computational chemistry and molecular modeling tool. LigPrep was used to desalt and then generate all possible conformations at pH 7.0 using Epik, and retain the specified chiralities. Finally, the OPLS_2005 force field was selected to minimize different compounds. Protein Preparation Wizard was used to optimize receptors. Glide scoring was used to assess the affinity between the CSPs and fenpropidin enantiomers.

### 3.7. Sample Preparation

Soil samples were gathered from a cropland in Nanjing in 0–15 cm depth, mixed, air-dried and, finally, sifted the soil with a 2 mm screen. Bananas, grapes, apples, wheat, soybeans and rice were all bought from local supermarkets in Nanjing. There was no substance in the Rt of fenpropidin. All samples were homogenized and stored at −20 °C. Fenpropidin was not found in any of the matrices.

Matrix samples were extracted according to the improved QuEChERS. Each sample was weighed 5 g using an electronic balance and then transferred into a 50 mL polytetra-fluoroethylene centrifuge tube. After wetting the soil with 5 mL ultrapure water, 10 mL acetonitrile was added. Other matrices only contained 10 mL acetonitrile. Then, the mixture was vortexed for 5 min at 2500 rpm, and fully extracted by ultrasound for 13 min. After completing the above steps, 3 g NaCl was added to the centrifuge tube followed by vortexing for 2 min, and then centrifugation at 4000 rpm for 5 min. After centrifugation, 1.5 mL of the organic phase was shifted to a centrifuge tube containing 50 mg PSA for purification. Then, it was violently shaken for 1 min and centrifuged at 8000 rpm for 5 min. The supernatant was filtered by a 0.22 μm nylon syringe filter; then, using UPLC-MS/MS, the samples were analyzed.

### 3.8. Method Validation

The specificity, ME, linearity, LOQ, accuracy and precision of the method were assessed based on the SANTE/12682/2019.

The blank samples were determined to observe whether there were interfering compounds near the Rt of the fenpropidin enantiomers. The solvent standard curve and matrix matching calibration curve were carried out to calculate the linearity and ME. The ME was evaluated by Equation (10). The LOQ was taken for the lowest validated spike level that met the requirement of the average recovery (70–120%) and RSDs ≤ 20% in each matrix. The repeatability and reproducibility were assessed on the same day and three nonconsecutive days through five replicates of six spiked samples (apples, grapes, bananas, soybean, wheat and rice) at three different levels (5–500 μg/kg) and soil at four different levels (5–1000 μg/kg). The average recoveries and the RSDs were calculated to analyze the accuracy and precision.
(10)ME=(the slope of matrix matched curve−the slope of solvent curve)/the slope of solvent curve×100%
the |ME| values between 20% and 50% have moderate ME, more than 50% are considered as strong ME, less than 20% shows no ME.

### 3.9. Stability of Fenpropidin Enatiomers

The fenpropidin enantiomers were dissolved in the solvents (acetonitrile, methanol and water) with a concentration of 0.5 mg/kg and kept at 4 °C and 25 °C. Then the concentrations of fenpropidin enantiomers at 0, 1, 3, 7, 14, 30, 60, 120 and 180 days were detected by UPLC-MS/MS under the optimal separation conditions to study the stability.

### 3.10. Enantioselective Dissipation in Soil

Soil dissipation experiments were carried out in Nanjing, China, and the soil without fenpropidin was selected as the test field. Three treatment plots and one control plot with 10 m^2^ of each plot were set, and each test plot was isolated by a 1 m wide isolation belt. A 10% fenpropidin emulsion solution was sprayed in the treatment plot at 1000 g (a.i.)/ha. Soil samples were collected at 2 h, 1, 3, 5, 14, 21, 28 and 35 d, and the samples were evenly mixed before being stored at −20 °C.

Enantio-selective dissipation of the fenpropidin enantiomers in soil was assessed by the first-order kinetic equation (Equation (11)), the half-life (t_1/2_) and enantiomer fraction (EF) were obtained by Equations (12) and (13).
(11)$$C_{t}=C_{0}e^{-kt}$$
(12)t1/2=ln2/k=0.693/k
(13)$$EF=C_{S}/(C_{S}+C_{R})$$
where C_0_ and C_t_ are the concentration at time 0 and t; k stands for the dissipation rate constant. C_S_ and C_R_ are the concentration of the *S*- and *R*-fenpropidin, respectively.

## 4. Conclusions

In this study, a fast and trace chiral analytical method for the enantioselective determination of the fenpropidin enantiomers in food and environmental samples using UPLC-MS/MS combined with the Lux cellulose-3 was established. The best enantio-separation conditions of the fenpropidin enantiomers were acquired by the single factor analysis combined with a surface response method. The absolute configuration of two fenpropidin enantiomers was confirmed by specific rotation and calculated and experimental ECD spectra. The elution order was *S*-(-)-fenpropidin and *R*-(+)-fenpropidin on the Lux Cellulose-3. The molecular docking results revealed the enantio-separation mechanism of the fenpropidin enantiomers. The excellent linearity, accuracy and precision acquired in the seven food and environmental matrix showed that the method was reliable. *R*-(+)-fenpropidin dissipated faster than *S*-(-)-fenpropidin. The establishment of the chiral analysis method breaks the limitation of traditional analysis of fenpropidin at the racemic level, which can more accurately monitor the residual of the fenpropidin enantiomers in the environment and food, and provide a basis for environmental risk assessment.

## Figures and Tables

**Figure 1 molecules-27-06530-f001:**
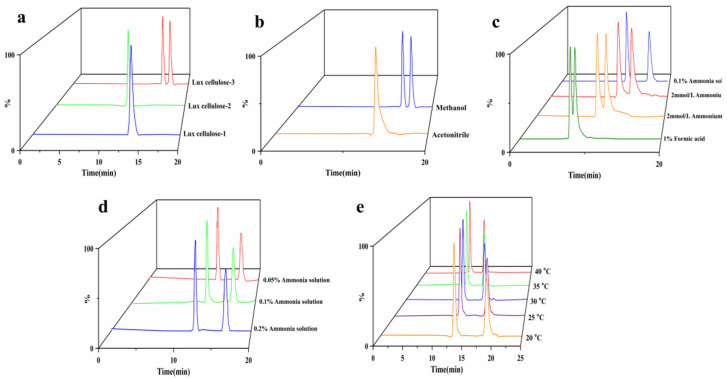
Chromatograms under different stationary phases (**a**), organic phases (**b**), aqueous phases (**c**), ammonia solution proportion (**d**) and temperature (**e**).

**Figure 2 molecules-27-06530-f002:**
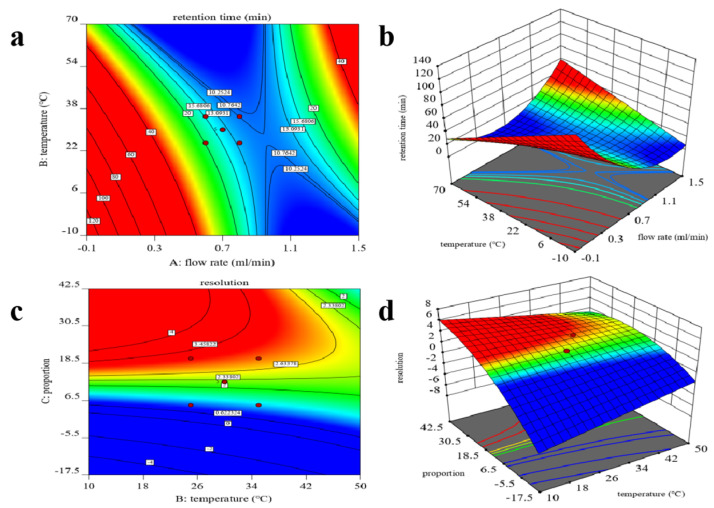
Two-dimensional contour plot (**a**,**c**) and three-dimensional response surface plot (**b**,**d**) for the influence of the flow rate, temperature and mobile phase ratio on Rt and Rs.

**Figure 3 molecules-27-06530-f003:**
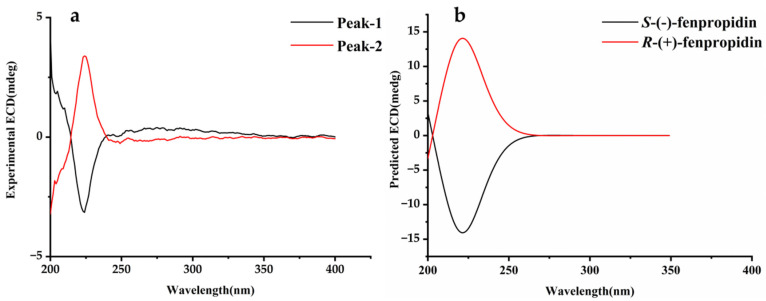
Experimental (**a**) and calculated ECD spectrum (**b**) of fenpropidin enantiomers.

**Figure 4 molecules-27-06530-f004:**
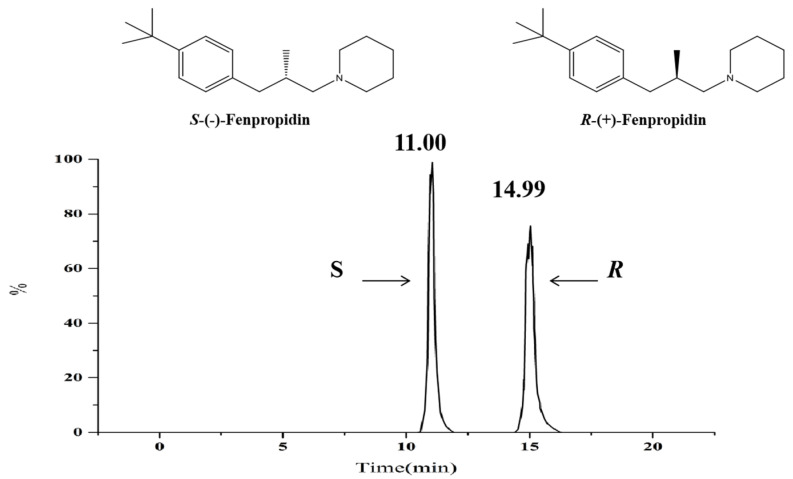
The chemical structures and typical chromatograms of the fenpropidin enantiomers.

**Figure 5 molecules-27-06530-f005:**
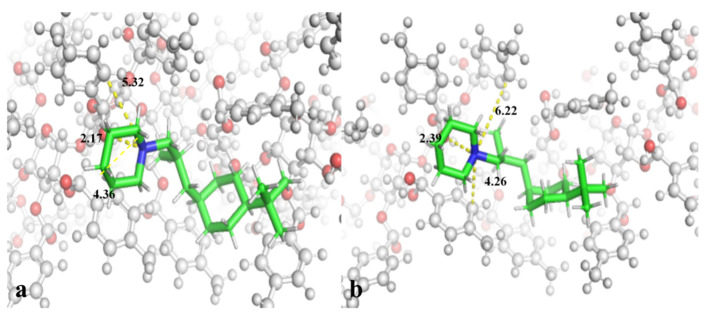
The docking posture of *R*-fenpropidin (**a**) and *S*-fenpropidin (**b**) with L3.

**Table 1 molecules-27-06530-t001:** Linear regression equation, matrix effect and LOQs for the fenpropidin enantiomers in different matrices.

Compounds	Matrix	Regression Equation	R^2^	Matrix Effect (%)	LOQ (μg/kg)
*S-*(-)-fenpropidin	acetonitrile	y = 10537x − 32578	0.9998		
	wheat	y = 10501x + 43144	0.9991	−0.26	5
	grape	y = 8249.5x + 19379	0.9999	−21.71	5
	apple	y = 9070.7x − 16612	1.s0000	−13.92	5
	banana	y = 7968x − 6797.5	1.0000	−24.38	5
	soybean	y = 10552x + 30602	0.9995	0.14	5
	rice	y = 8508.9x + 16128	0.9997	−19.25	5
	soil	y = 10666x + 43568	0.9989	1.22	5
*R-*(+)-fenpropidin	acetonitrile	y = 11301x − 59712	0.9995		
	wheat	y = 9972.5x + 68949	0.9988	−11.76	5
	grape	y = 8346.9x + 9890.1	1.0000	−26.14	5
	apple	y = 9404.4x − 24369	0.9999	−16.78	5
	banana	y = 8559.4x − 16592	1.0000	−24.26	5
	soybean	y = 10378x + 29535	0.9995	−8.17	5
	rice	y = 8657.3x + 2716.4	0.9998	−23.39	5
	soil	y = 11004x + 19222	0.9993	−2.63	5

**Table 2 molecules-27-06530-t002:** Accuracy and precision of the method in the seven matrixs.

Compounds	Matrix	Spiked Level (μg/kg)	Intraday (*n* = 5)						Interday (*n* = 15)
			Day 1		Day 2		Day 3		RSD (%)
			Mean Recovery (%)	RSD (%)	Mean Recovery (%)	RSD (%)	Mean Recovery (%)	RSD (%)	
*S*-(-)-fenpropidin	wheat	5	89.3	1.3	89.3	1.2	90.1	1.3	1.2
		50	98.1	4.7	102.8	3.6	103.7	2.7	5.1
		500	91.6	2.5	91.1	3.4	90.9	3.3	2.6
	grape	5	91.9	4.2	94.8	3.6	97.7	1.4	4.6
		50	92.5	1.6	93.3	3.0	95.2	2.9	2.7
		500	90.1	4.7	89.6	4.6	92.0	2.3	3.5
	apple	5	101.3	3.4	104.0	2.8	103.4	2.6	3.2
		50	91.7	3.1	89.8	1.3	89.7	1.2	2.4
		500	100.7	8.8	100.8	8.5	96.7	8.9	8.0
	banana	5	83.2	4.0	80.7	3.0	78.6	0.7	3.9
		50	92.2	2.6	91.4	1.6	93.0	1.1	2.1
		500	94.5	0.7	92.9	3.1	90.4	4.5	3.8
	soybean	5	73.3	2.2	74.4	1.7	74.8	1.1	1.7
		50	71.5	0.3	71.9	0.5	71.9	0.6	0.5
		500	77.2	4.4	76.7	5.1	75.7	5.4	4.1
	rice	5	83.4	2.6	80.7	3.1	80.6	3.1	3.2
		50	83.3	4.8	80.8	0.6	80.2	1.0	3.8
		500	91.3	1.7	91.1	1.6	90.5	1.2	1.3
	soil	5	85.8	1.8	85.5	1.9	82.9	2.6	2.8
		50	86.3	1.8	86.3	1.8	87.4	1.4	1.6
		500	88.3	4.2	87.2	3.8	86.0	5.6	4.7
		1000	89.2	2.3	88.5	2.1	87.1	1.2	1.2
*R*-(+)-fenpropidin	wheat	5	95.9	4.4	93.2	4.0	94.9	3.7	3.8
		50	96.1	6.5	103.1	5.9	104.9	4.9	7.7
		500	94.0	2.2	92.8	3.8	92.3	3.8	2.7
	grape	5	89.4	2.8	90.4	2.4	92.9	2.2	3.1
		50	92.7	1.2	93.6	2.1	95.4	3.7	2.9
		500	90.3	6.1	91.5	5.8	94.7	1.1	4.4
	apple	5	97.8	0.4	99.2	2.1	99.1	2.1	1.7
		50	95.1	1.9	94.9	1.5	95.4	1.5	1.4
		500	94.1	2.0	93.7	2.7	92.2	2.6	2.3
	banana	5	103.8	4.5	106.1	0.7	104.2	3.4	3.8
		50	94.0	4.7	91.5	2.4	90.9	3.2	4.2
		500	96.4	4.3	97.6	4.0	100.0	1.3	3.4
	soybean	5	72.6	1.2	72.4	0.9	73.3	0.6	1.0
		50	71.7	1.2	71.6	0.7	72.2	0.9	0.8
		500	75.3	3.8	74.9	3.5	73.8	3.7	2.8
	rice	5	78.6	5.8	75.6	1.1	75.8	1.1	3.7
		50	81.5	3.3	80.4	1.8	78.3	2.5	3.1
		500	91.1	1.6	90.8	1.2	90.2	1.1	1.2
	soil	5	92.2	5.7	95.7	5.1	94.9	4.7	4.8
		50	88.0	3.1	86.2	1.7	88.1	3.0	2.9
		500	87.8	4.5	87.3	4.0	86.5	5.2	4.1
		1000	90.2	2.1	87.1	2.3	88.4	1.3	3.8

## Data Availability

Appendix A is available.

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
