# Peer review of "An Innovative Chiral UPLC-MS/MS Method for Enantioselective Determination and Dissipation in Soil of Fenpropidin Enantiomers"

_molecules, 2022, doi:10.3390/molecules27196530_

Round 1

Reviewer 1 Report

1. The “2. Results and Discussion” and “3. Materials and method” should be changed so that readers can understand the experimental process and make the article easier to understand.

2. The all equation should be supported by literature.

3. Page 1, Line 39-40. You need to use “Lemna minor” instead of Lemna minor.

4. Page 1, Line 50-52. “ The mode action of fenpropidin is inhibition to Δ14 - reductase and Δ8 → Δ7 - isomerase in powdery mildews and other fruits and crops.” This sentence is incorrectly expressed. Please rewrite it.

5. Page 2, Line 87-96. Has the effect of different mobile phases on separation of fenpropidin enantiomers on different chiral stationary phases been studied?

6. Page 2, Line 89. “CSP is the dominant factors in chiral recognition.” This sentence should be “CSP is the dominant factor in chiral recognition.”

7. Page 3, Line 101. Does Rt stand for retention time or retention?

8. Page 6, “The distance ≤ 6 Å and an angle 60° ≤ θ ≤ 90° between the pi system and the cation center meet the Geometric criteria.” In the cited literature, this sentence describes the characteristics of the cation-π interaction rather than the π-cation interaction. These are two different forces. Thus, this sentence should be deleted.

9. According to the cited literature, it is suggested to change pi-cation to π-cation.

10. Unify units throughout the text, such as LOQ units in line 231 and Table 1.

11.  Some data of recovery should be described by mean ± SD, such as ME in the table 1 and mean recovery in the table 2.

12. In the line 235, “There was no ME in rice with S-(-)-fenpropidin, but there was matrix suppression effect in rice with R-(+)-fenpropidin.” The meaning of this sentence is confusing and the description does not match the data given in Table 1.

13. Page 7, Line 234. Does Matrix Effect (ME) need to be capitalize?

14. I don't see the experimental steps about 2.6 Stability of Fenpropidin Enatiomers.

15. In the line 397, the 10 m2 should be changed to 10 m2.

Reviewer 2 Report

In this manuscript, Wang's group has demonstrated the new UPLC-MS based method for the determination of enantiomers in different food items and soil. The author also carried out a molecular docking study to study the chiral mode of action. The author thoroughly investigated and discussed the different parameters affecting the resolution of two enantiomers and showed Lux cellulose column gives the best separation when methanol-ammonia is used as a mobile phase.

The manuscript is clearly written and easy to understand and the author has cited proper references. In many places articles (a, an and the) are missing.

In my view, the article is suitable for publication in the journal “Molecule” with minor grammatical corrections (use of articles).
